

# A novel techno-economical layout optimization tool for floating wind farm design

Amalia Ida Hietanen[1], Thor Heine Snedker[1], Katherine Dykes[2], and Ilmas Bayati[1]

[1]PEAK Wind, Jens Baggesensvej 90K st, 8200 Aarhus N, Denmark
[2]DTU, Frederiksborgvej 399, 115, S20, 4000 Roskilde, Denmark

**Correspondence:** Amalia Ida Hietanen (ahi@peak-wind.com)
Advanced Programs PEAK Wind (advanced.programs@peak-wind.com)

**Abstract.** Over the past few years, the offshore wind sector has been subject to a renewed yet growing interest from the industry and from the research sphere, with a particular focus on a recently developed concept being the Floating Offshore Wind (FOW). Because of its novelty, floating research material is found in limited quantity. This paper focuses on the layout optimization of a Floating Offshore Wind Farm (FOWF) considering multiple parameters and engineering constraints, combining floating
specific parameters together with economic indicators. Today's common wind farm layout optimization codes do not take into account neither floating specific technical parameters (anchors, mooring lines, inter-array cables (IAC) etc) nor non-technical parameters (OPEX, CAPEX, other techno-economic project parameters). In this paper, a multi-parametric objective function is used in the optimization of the layout of a FOWF, combining the AEP together with the costs that depend on the layout. The mooring system and the collection system including the inter-array cables and the offshore substation are identified as
layout-dependent and therefore modelled in the optimization loop. Using ScotWind site 10 as a study case, it was found with the predefined technical and economic assumptions that the profit was increased by 34.5 m euros compared to a grid-based layout. The main drivers were identified to be the AEP, followed by the anchors and the availability associated to the failures of inter-array cables.

## 1 Introduction

Today, offshore wind farms are located for the majority in shallow water areas, where it is possible to install Bottom-fixed Offshore Wind Turbines (BOWT). Monopiles remain the preferred foundation choice of developers with over 80% of all installations in 2020, and jacket structures coming in second with 10% of the installations (Ramírez et al., 2020). While the first full-scale FOWF was installed in 2017 off the coast of Scotland, FOW still appears as the next frontier to cross in the wind industry. Nevertheless, Floating Offshore Wind Turbines (FOWTs) come with a state-of-the-art technology, that allows
to exploit areas with a water depth above 60 m, which is unfeasible for BOWTs. Indeed, FOWTs differ from BOWTs as they are not fixed to the seabed on a foundation, but attached with a mooring system. Hence, FOW appears as a solution to harness the full potential of offshore wind while reducing the constraints in terms of water depths and soil conditions.

However, to be economically competitive with Bottomo-fixed Offshore Wind Farms (BOWF), the costs of FOWF projects need to be minimized to make them more attractive for developers and investors. Due to their complex and novel technol-





ogy, FOWTs have higher installation, maintenance and decommissioning costs than BOWTs. The main reason for that is the
limited site accessibility because of possible incompatible weather conditions, expensive installation procedures and high grid
connection costs. The CAPEX of FOWTs end up being about twice the CAPEX of BOWTs (Maienza et al., 2020). FOW cost
reduction is therefore an area that needs to be investigated - for example through layout optimization, which is the focus of this
project.

## 2   Methodology

### 2.1   Problem formulation

In this section, the FOWF layout optimization framework is presented. The problem takes the form of a multi-constrained and
multi-parametric maximization problem as given in Eq. (1).

$$\max_{\omega \in \Omega} \mathbb{J}(\omega) \tag{1}$$

where $\omega$ stands for the different decision variables, $\Omega$ is the set of constraints applied to $\omega$, and $\mathbb{J}$ is the cost function which
derives from the project's relative Net Present Value (NPV).

#### 2.1.1   Assumptions

Below, a list of the assumptions considered in the problem is given.

- The number of FOWTs in the wind farm is fixed to $N$.

– All FOWTs are assumed to be identical, meaning that the rotor diameter, the hub height, the rated power and the cut-in
and cut-out wind speeds are the same across the whole wind farm.

- The area $\mathcal{A}$ of the wind farm is fixed, and real wind resource and sea-bed data specific to the chosen site are used in the
modelling.

- A uniform sea-depth $z_{depth}$ is calculated as the average depth of the site area $\mathcal{A}$.

– The wind resource is spatially uniform.

#### 2.1.2   Decision variables

The set of design variables of the framework is chosen to be the coordinates $(x_i, y_i)_{i \in \{1,2,...,N\}}$ of the FOWT centroids, as
presented in Eq. (2).

$$\omega = [\omega_1, \omega_2, \ldots, \omega_N] \quad \text{with} \quad \omega_i = [x_i, y_i], \quad i \in \{1, \ldots, N\} \tag{2}$$





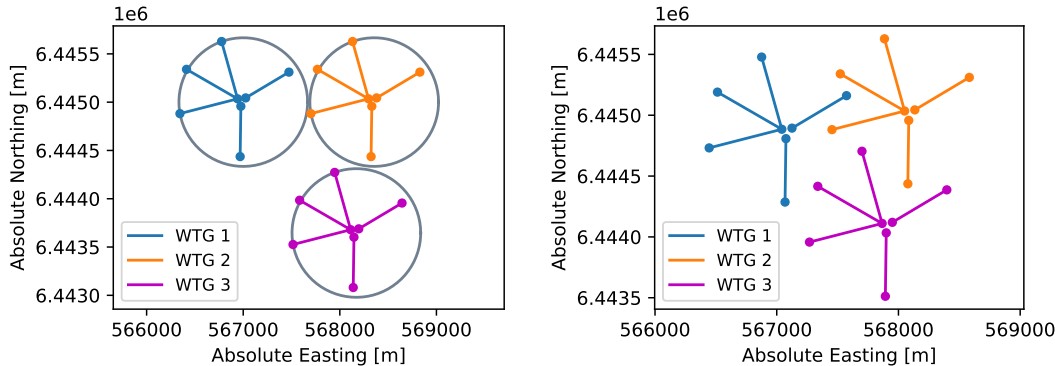

**Figure 1.** Possible layouts for three turbines using the circular constraint where the mooring lines are constrained in circles (left) and geometrical constraint where the mooring lines can be overlapping (right)

### 2.1.3 Constraints

The design variables are subject to engineering and operational constraints, defined in Eq. (3).

$$\Omega = \begin{cases} (x_i, y_i)_{i \in \{1,2,...,N\}} \in \mathcal{A} \\ \sqrt{(x_i - x_j)^2 + (y_i - y_j)^2} > d_{min} \\ \min(dist(p_i, p_j)) > d_{min}^{mooring} \quad \forall (i,j) \in [1,N]^2, i \neq j \end{cases} \tag{3}$$

– The first constraint means that all the FOWTs and their mooring system fall inside of the site area $\mathcal{A}$.

– The second constraint sets a minimum distance $d_{min} = 2.95D$ between the centroids of the FOWTs - which is equal to the footprint of one mooring line.

– The last constraint sets a minimum distance between any point $p_i$ of the mooring lines of the turbine $i$ and any point $p_j$ of the mooring lines of the turbine $j$.

In the first constraint, it is necessary to make sure that all components of the FOWTs are inside of the area $\mathcal{A}$, especially in the case of floating. In traditional layout optimization problems, the wind turbines are BOWTs, which doesn't require much more than constraining the centroids of the BOWT in the site area. In this project, to account for the anchors and mooring lines, a buffer zone that reduces the site by the footprint of the mooring lines is constructed. Therefore, the area $\mathcal{A}$ is defined as the area inside of the buffer zone.

The mooring distance constraint allows to give more freedom in the layout design. Indeed, if a circular distance constraint is considered to account for the mooring lines, then it reduces the layout possibilities in comparison to the constraint chosen in this project, where the mooring lines can overlap while respecting a limit distance as it is shown in Fig. 1.

The two distance constraints could in theory be combined because if the mooring distance constraint is satisfied then the centroid distance would be satisfied as well. However, using only the constraint on the mooring lines, though it is directly



related to the set of coordinates $(x_i, y_i)_{i \in \{1,2,...,N\}}$, it can lead to a scenario within the optimization where two turbines are at the exact same position. The latter would lead to an error and the optimization would break. $d_{min}$ is therefore the very minimal distance that two turbines can be separated by, i.e. the footprint length of one mooring line.

### 2.1.4 Objective function

The objective function chosen in this project derives from the NPV, which is the total profit of the wind farm through its lifetime, converted to present day value. It is a scalar-valued cost function that includes the AEP, the price of electricity $p_{kWh}$, the OPEX and the CAPEX. It only includes the components $Comp(\boldsymbol{X})$ that depend on the turbines positions, and not the fixed cost components $Comp_{fixed}$. As it has been stated by Tesauro et al. (2012), the costs that are not influenced by the actual wind farm layout (cost of planning, cost of the civil infrastructure, price of the electrical connection to the main grid etc.) are considered irrelevant and not modeled in the project framework. These fixed costs can be added as a post-processing calculation, as they are not related to the layout.

The objective function used in the project is given in Eq. (4).

$$Obj = \sum_{i}^{N} \left( AEP(x_i, y_i) p_{\text{kWh}} - OPEX(x_i, y_i) \right) a - CAPEX(x_i, y_i) \qquad (4)$$

where $AEP(x_i, y_i)$ is the Annual Energy Production including layout variable losses, $p_{kWh}$ is the price of electricity, $OPEX(x_i, y_i)$ and $CAPEX(x_i, y_i)$ are the layout-variable components of the OPEX and of the CAPEX respectively, and $a$ is an annuity factor defined in Eq. (5).

$$a = \frac{1 - (1 - r)^{-n_y}}{r} \qquad (5)$$

where $r$ is the interest rate and $n_y$ the lifetime of the wind farm.

In this project, the NPV was preferred over the Levelized Cost Of Energy (LCOE) because the LCOE has the drawback of increasing, i.e. get worse, as the size of the wind farm increases. On the other hand, the total profit of a project would typically increase with the size of the wind farm (EMD (2022)). On top of that, the time dependency of the price of electricity and of the OPEX in the NPV, can later be included in the optimization tool to assess of the evolution of the NPV over the lifetime of the FOWF. That being said, both the LCOE and the NPV are relevant to use in an optimization framework, as they both have their own advantages and drawbacks.





## 2.2 Summary of the Techno-Economic modelling

| Category | Parameters | Description |
|---|---|---|
| **Power production** | $AEP_{pot}$ | AEP including wake losses |
| | $\eta_{elec}$ | Electrical losses in the IAC |
| | $\eta_{avail}$ | Availability losses due to the failure of IAC |
| **CAPEX** | $C_{cables}$ | Cost of IAC |
| | $C_{anchors}$ | Total cost of anchors |
| **OPEX** | $C_{IAC,failure}$ | Cost of IAC to replace due to failure |

**Table 1.** Summary table of the objective function's components

Therefore, the final objective function is summarized in Eq. (6).

$$Obj = \sum_i^N ((1 - \eta_{tot}(x_i, y_i))\, AEP_{pot}(x_i, y_i) p_{kWh} - C_{IAC,failure}(x_i, y_i))a - C_{anchors}(x_i, y_i) - C_{cables}(x_i, y_i) \tag{6}$$

where $\eta_{tot}$ is the efficiency of the layout-variable production losses defined in Eq. (7) and $AEP_{pot}$ the potential AEP.

$$\eta_{tot} = 1 - \prod_{i=1}^{N_{losses}} (1 - \eta_i) = 1 - \prod_{i=1}^{N_{losses}} (1 - \frac{AEP_{loss}^i}{AEP_{pot}}) \tag{7}$$

## 2.3 Optimization algorithm

In this work, a gradient-free heuristic optimizer based on a Random-search algorithm by Feng and Shen (2015) is used. This algorithm was developed in Python within the package TopFarm - based on the OpenMDAO library. The random-search
algorithm starts from an initial feasible layout and then improves it iteratively.

A random search algorithm was chosen over a gradient-based algorithm because it allows to move out of the local optima. Gradient-based optimization algorithms tend to quickly converge towards local optima, especially when the initial layout is close to a local optima. On the contrary, random search presents the advantage of never getting stuck in local optima since the positions of the turbines are moved randomly. To make sure that the random search provides the best solution, simulations can
be run several times, to obtain a distribution of the optimal results.

## 2.4 Modelling of floating specific components

### 2.4.1 Floating platform

The floating platform is modelled using the geometry of a semi-submersible floating platform, with 5 mooring lines. The floating platform orientation $\phi$ - which also drives the orientation of the mooring lines - is aligned to the main wave direction,
to maximize the stability of the structure.





### 2.4.2 Mooring lines

The mooring lines are attached to the fair-leads of the floater, positioned at each of the corners of the platform. They are then anchored to the seabed. The mooring system gives some freedom to the FOWTs to move their positions laterally through surge and sway motions. Following in the steps of the oil and gas industry, FOWTs platforms are designed together with their

mooring system in order to reduce the lateral displacements (Mahfouz et al., 2022). The floating platform in this project has an asymmetric mooring system (3 mooring lines in the "left" part, and two mooring lines in the "right" part of the platform) which means that the FOWTs will have different distances relative to each other for each wind direction. Therefore, to avoid collision and friction and to facilitate operations, the mooring lines must be separated by a certain distance, and most importantly the mooring lines are not allowed to cross.

Therefore, a distance constraint between the mooring lines is implemented to avoid this problem. This constraint is included within the optimizer by calculating at each iteration the distances between the different mooring lines of each turbine. A distance matrix sized $N \times n_{mooring}$ is computed - with $n_{mooring}$ the number of mooring lines per FOWT - and if the minimum value of this matrix is greater than the distance constraint, then the distance constraint for mooring lines is satisfied and the layout can be retained. To reduce the computational effort, the strictly upper triangular matrix only is computed since the

distance matrix is symmetrical and has zeros on its diagonal.

### 2.4.3 Anchors

According to Lieng et al. (2022), manufacturing and installing anchoring systems is a major cost driver of a FOWF. The cost of anchors depends on the required holding power and weight, which are driven by the seafloor technical conditions. For instance, it is easier to install an anchor in sand than in bedrock (DTOcean, 2015).

Different types of anchors exist in the industry: drag-embedded, driven piles, suction piles, gravity anchors etc. In this paper, two types of anchors are considered: drag anchors for cohesive sediments (e.g. sand-based seabed) and driven anchors - applicable to a wide range of seabed conditions but much more difficult and expensive to install, and difficult to remove upon decommissioning (Ros and James (2015)). In the design of the wind farm, the idea is to use an optimal - but not necessarily minimal - number of drilled anchors to reduce costs, installation risks and environmental impacts.

To do so, a map of the seabed bedrock depth is included in the optimization. The zones where the bedrock is more than 5 m under the sand are defined as suitable for drag anchors, while the rest of the area needs drilled pile anchors. Therefore, a binary map is created as shown in Fig. 2, where the blue zones are sand-prevailing and the red zones are bedrock-prevailing. Eventually, the idea is to compute the positions of the anchors at each iteration in the optimizer, and to compute the associated costs according to the type of seabed where the anchors fall. Further, this anchor cost component is included in the objective

function, and the optimizer evaluates how valuable it is to move a FOWT from a bedrock-prevailing zone to a sand-prevailing zone, according to its contribution to the overall costs.





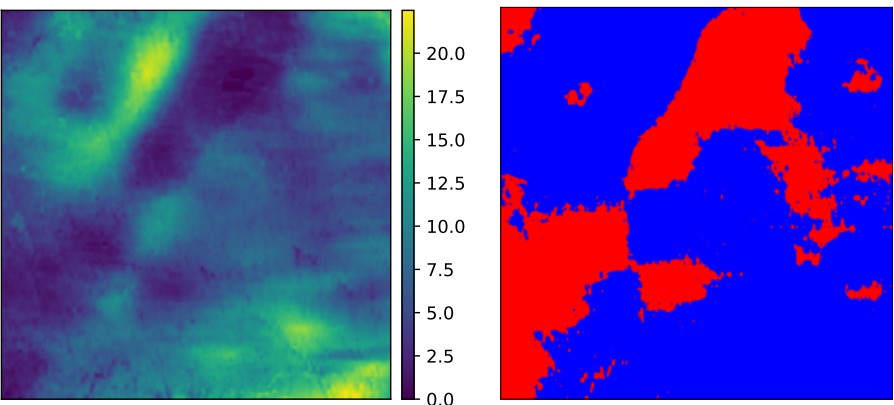

**Figure 2.** Non-binarized (left) and binarized (right) maps of the bedrock depth under the seafloor

*In the model, it is possible to mix the kinds of anchors on a FOWT, meaning that a given FOWT can have both drilled anchors and drag-embedment anchors.*

**2.4.4 Dynamic cables**

Inter-array cables are modelled and optimized in a sub-optimization routine that minimizes the total cable length, inside of the main optimization loop. The umbilical dynamic cable section is modelled as well to adjust the total inter-array cable length. Further, the electrical losses together with the availability losses due to cable failure are computed to correct the AEP.

– ***Cable routing*** The collection grid, and especially the inter-array cable layout, is highly dependent on the wind farm
layout. Not only does it affect the costs but it also plays a key role in the energy yield. It is therefore relevant to include the cable-routing design in the optimization loop, to evaluate its influence on the overall costs. To maximize the efficiency of the whole optimization, the cable layout is optimized by a sub-optimization algorithm at each iteration of the main optimizer.

The cable routing optimization is based on the Esau-Williams heuristic algorithm and was developed by Souza de Alencar
(2022). The objective function of the algorithm is to minimize the total cable length. The algorithm finds sub-optimal solutions that are very close to the exact solutions, but it produces on average better results than many other heuristics. This algorithm has a very good performance and accuracy with a low computational effort, which is crucial here since an optimized cable layout needs to be computed at each iteration of the main optimization. Because the optimization already has a great complexity with multiple constraints and parameters, the cable optimization is kept simple. The
Esau-Williams heuristic is built using a minimal cost spanning tree of a graph, with designated roots, nodes and capacity constraints. In this paper, the roots of the spanning tree are the Offshore SubStation(s) (OSS) while the nodes are the FOWTs. The algorithm allows sub-branches on a given string, while respecting a maximum number of nodes per string. An example of an optimized cable routing is provided below in Fig. 3.





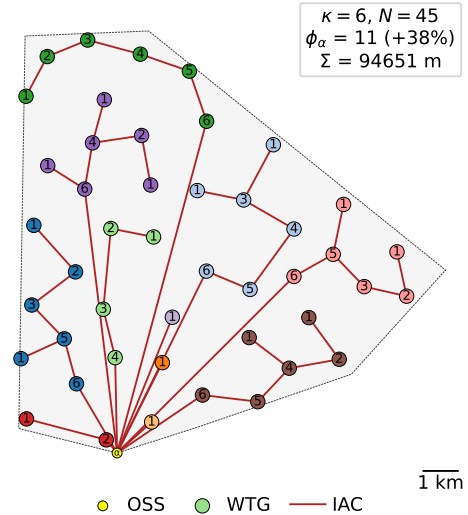

**Figure 3.** Optimized cable routing with one OSS and a max of 6 FOWT per cable string

Two constraints are defined in the cable sub-optimization algorithm:

– **Non-crossing constraint**: It prevents two IAC from crossing.

– **Capacity constraint**: It determines a maximum number of nodes per string. For cable layouts, this capacity constraint is defined by $\kappa_i$, i.e. the maximum number of turbines per cable string $S_i$, which is computed in Eq. (8).

$$\kappa_i = \frac{P_{i,capacity}}{P_{rated}} \quad \forall S_i \in \mathcal{S} \tag{8}$$

where $P_{i,capacity}$ is the capacity of the cable $S_i$, $P_{rated}$ is the rated power of a turbine and $\mathcal{S}$ is the set of cable strings.

*The power capacity of a cable is defined by the core-size of the cable. In practice, inter-array cable layouts have 2 or 3 core-sizes with smaller sizes at the end of the strings to reduce the costs. In this project, to simplify the process, a single core-size is used.*

– ***Dynamic section*** While BOWTs' inter-array cables are installed buried or secured on the seabed, FOWTs' inter-array cables have a dynamic section that enables them to move together with the floating platform. Dynamic sections can be either in catenary shapes or in lazy-wave umbilical shape. It was shown by Rentschler (2020) that the catenary shape is not suited for water depths above 100m while umbilical shapes can be used in water depths of more than 200m. On top of that, catenary shapes are more susceptible to platform movements than umbilical shapes, and are therefore prone to higher compression and fatigue at the touchdown point. On the other side, umbilical shapes decouple platform and cable movement which makes them preferable over catenary shapes.





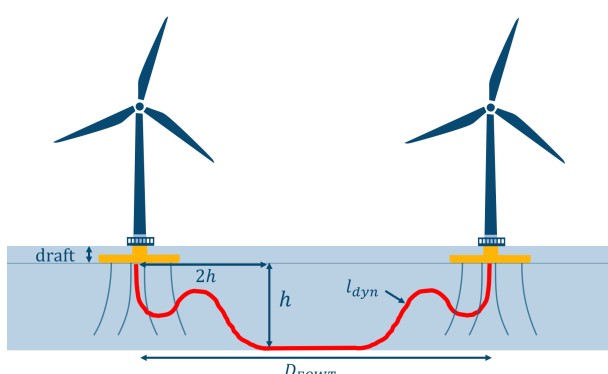

**Figure 4.** Dynamic cable in lazy wave shape

The cable is fixed at a certain distance from the water surface, corresponding to the draft of the floating platform. The hang-off over the seabed $h$ is given by Eq. (9). The horizontal distance of the cable's fixation point on the seabed is set to 2h, as recommended by Rentschler (2019).

$$h = z_{depth} - draft \tag{9}$$

Rentschler (2019) found a general design rule being that there is a constant ratio between the total length of the dynamic cable $l_{dyn}$ and $h$, as written in Eq. (10). This is retained as a general formula to compute the length of the Inter-Array Cables.

$$\frac{l_{dyn}}{h} \approx 2.782 \tag{10}$$

Eventually, the total cable length $L_{tot}$ is computed following (11), adjusted from Lerch et al. (2021).

$$L_{tot} = 1.05 D_{FOWTs} + 2N(l_{dyn} - 2h) \tag{11}$$

where $D_{FOWTs}$ is the horizontal total distance between the turbines connected together, $N$ is the number of FOWTs, $l_{dyn}$ is the dynamic section length and $2h$ is the horizontal distance from the two cable fixation points.

– **Offshore Substation** The OSS transmits the power from all the FOWTs to the shore transmission network, through an export cable towards an Onshore SubStation (OnSS). In this paper, the number of OSS and their positions are fixed, so the length of the export cable(s) as well as the associated losses will be fixed. Therefore, the costs associated to the OSS, to the export cables, to the OnSS are not included in the optimization loop. Since the OSSs in a FOWF are also generally mounted on floating structures, the dynamic cable sections of the cables connected to the OSS are computed as written in Eq. (10) and added to the total cable length (11).



### 2.4.5 Electrical losses

For the cable section $S_i$, the power loss $P_{loss}^i$ is given by Eq. (12).

$$P_{loss}^i = 3\left(\frac{P_{gen}^i + P_{trans}^i}{\sqrt{3}U}\right)^2 R_{cable}^i L_{cable}^i \quad \forall S_i \in \mathcal{S} \tag{12}$$

where $P_{gen}^i$ is the power generated by $FOWT_i$ at the end of the cable $S_i$, $P_{trans}^i$ is the power transmitted to $FOWT_i$, $U$ is the voltage applied, $R_{cable}^i$ is the resistance of cable $S_i$ and $L_{cable}^i$ is the length of cable $S_i$.

The power losses are computed starting from the end of a cable string where no power is transmitted from a downstream
turbine, and then the losses are computed at each upstream FOWT until the OSS is reached. Eventually, the final power losses
$P_{loss}^{elec}$ are given by Eq. (13).

$$P_{loss}^{elec} = \sum_{i \in [1,N]} P_{gen}^i - \sum_{n_{OSS}} P_{trans}^{OSS} \tag{13}$$

where $n_{OSS}$ is the number of OSS and $P_{trans}^{OSS}$ is the power transmitted to the OSS.

While the electrical power loss is linearly linked to the length of the cable, it is proportional to the square of the power going
through the cable. The power loss is also closely linked to the cable layout, and especially:

– The number of cable strings in the cable layout

– The number of turbines per cable string

– The length of the cable sections

From an analysis carried out in this study on the cable routing subroutine - using a wind farm with an AEP of 2550 GWh, and
with 45 FOWTs - it is shown in Fig. 5 that the number of cable strings drives the electrical losses. If all turbines are connected
on a single string, then the production loss explodes and reaches over 6% of the AEP, while if the layout contains more strings,
the production loss drops. With 45 cable strings, the electrical loss is below 0.2%. However, the length of the cables increases
with the number of cable strings. Therefore, it is important to combine the production together with its losses and the material
costs to find a balance and reach the optimum in the optimization phase.

Figure 5 is obtained from the Esau-Williams heuristic cable optimization. The number of cable strings is computed internally
within the optimizer. The only input here is the maximum number of turbines per string, which affects the number of strings.
This is the reason why there are gaps between $n_{strings} \in [17,27]$ and $n_{strings} \in [27,45]$, the optimizer creates strings of 1
FOWT, then strings of 2 FOWTs etc. However, even if it is allowed to create strings of 2 FOWT, the solver can find a better
solution with only 1 FOWT on a string. For example, the point at $n_{strings} = 27$ corresponds to 18 strings of 2 FOWTs and 9
strings of 1 FOWT. (See Fig. 6)





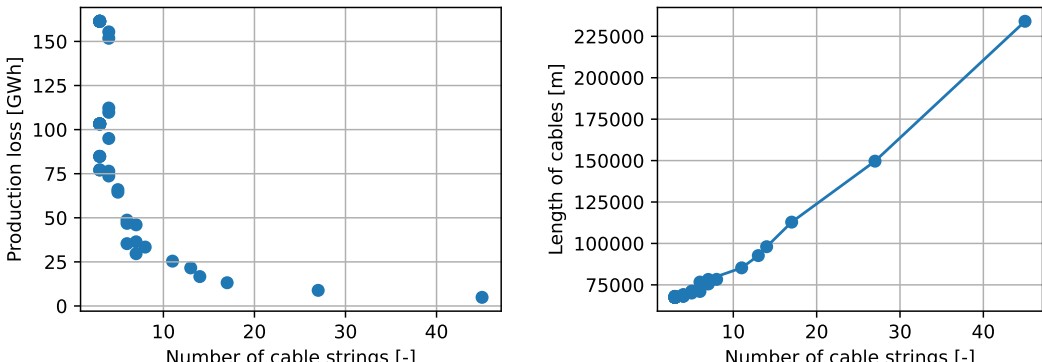

**Figure 5.** Number of cable strings as a function of the cable length (left) and Annual electrical losses as a function of the number of cable strings (right)

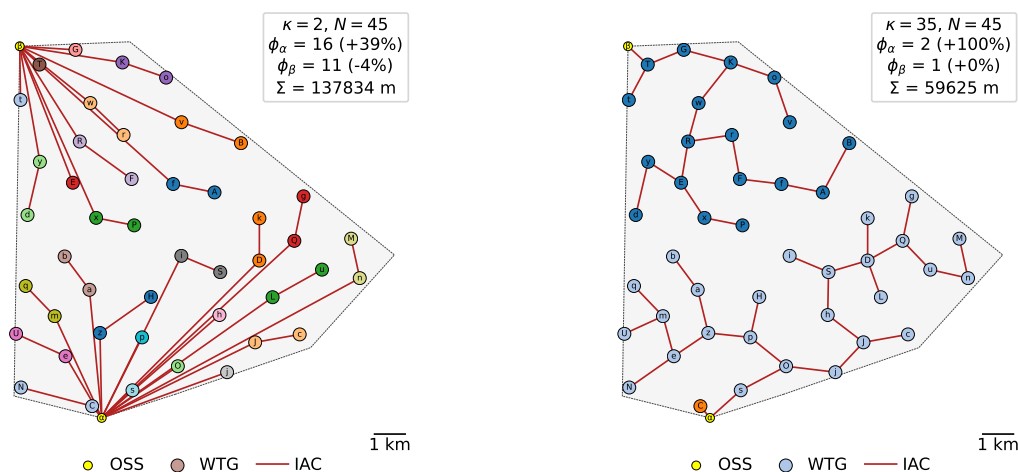

**Figure 6.** Cable routing for $\kappa = 2$ (left) and $\kappa = 35$ (right)

### 2.4.6 Availability losses

When it comes to wind farm layout optimization, the availability of the IAC comes as a layout-variable loss. Indeed, according to the cable layout, and especially how the turbines are organized in the cable routing (number of strings, number of turbines per string etc), a cable failure can potentially disrupt global production to a greater or lesser extent.

For example, Fig. 6 shows two cable routings, with maximum 2 FOWTs per string (left) and maximum 35 FOWTs per string (right). In the first case, the production of 1.4 FOWTs will be affected by a cable failure, on average. As for the second case, the production of 5.8 FOWTs will be affected by a cable failure on average.





To compute the availability losses, for each cable section, the power produced by the turbine at the end of the cable section together with the power produced from the downstream turbines are computed. Eventually, the losses $P_{loss}^{avail}$ triggered by a
failure of each of the cable sections are computed following Eq. (14).

$$P_{loss}^{avail} = \sum_{S_i \in \mathcal{S}} FR_i(P_{gen}^i + P_{trans}^i) \tag{14}$$

where $FR_i$ is the failure rate of the cable $Si$.

The failure rate for each of the cable section is computed following the research from Zhang et al. (2023) where they have investigated the failure rate of submarine cables and found that the failure rate is a function of the cable length. Their findings
are adjusted to be compliant with dynamic cables as shown in Eq. (15) using assumptions from Lerch et al. (2021).

$$FR_i = \begin{cases} 0.0094 & L_{cable}^i < 9.33 \text{ km} \\ 0.0037L_{cable}^i - 0.025 & L_{cable}^i \geq 9.33 \text{ km} \end{cases} \tag{15}$$

Using the failure rate of each of the cable sections, the cost of cables to replace in case of failure can be computed, as shown in Eq. (16). This cost is an OPEX component, it stands for the cost to pay every year for cable failures.

$$C_{IAC,failure} = \sum_{S_i \in \mathcal{S}} FR_i L_{cable}^i C_{IAC} \tag{16}$$

where $C_{IAC}$ is the cost of IAC per unit length.

## 3    Study case

Scotland has a long history in developing floating systems starting with oil and gas. The expertise in this sector was used as a strong heritage to get started in the floating wind sector. While the first floating wind project, Equinor's 30 MW Hywind Scotland started operating in 2017, followed by Kinkardine in 2021 and Pentland when completed, Scotland appears as a world
leader in the deployment of floating wind. On top of these 3 floating wind farms, in 2022, a total number of 14 floating wind projects have been approved in Scotland through the ScotWind leasing. These projects will benefit the Scottish businesses and community as well as providing a major boost to reach the UK 'Net Zero' state goal (Marcus (2022)). They will also add close to 18GW of commercial scale floating wind, making Scotland the largest floating wind market in the world.

### 3.1    Site under study

In this paper, the site that was chosen to perform the layout optimization on is the site 10 from ScotWind projects, being Broadshore, with a capacity of 500 MW. Broadshore was preferred over the other ScotWind sites due to its relatively small number of turbines compared to the others. Indeed, to test and perform large sensitivity analyses using the optimization with an increased complexity due to the new FOW features, it is necessary to have a reasonable number of design variables, being $2N$ with $N$ the number of turbines.





A real study case was chosen to study the relevance and the performance of the techno-economic layout optimization. All inputs are set to be as close as possible from the reality so that the optimization is studied under realistic conditions. A detailed list of the inputs is presented in the Appendix A.

On top of the site inputs listed in Appendix A, a bedrock map is required to calculate anchors costs. However, seafloor studies, including measurements of the depth of the seafloor bedrock, are typically carried out by oceanographers or marine geologists using specialized equipment and vessels, which is rather expensive. Therefore, seafloor properties maps are generally not publicly available, and detailed seafloor surveys are carried out only on request, in restrained zones. Hence, a map of the water depth is used instead.

## 3.2 Results

The optimization of the FOWF layout using the relative NPV as the objective function is run using the assumptions in Sect. 2.1.1 and the inputs in Appendix A. One must note that the result obtained is one of the possible optimal layouts - see Sect. 3.3 for further details.

To report on the performance of the techno-economic layout optimization, the optimized results are compared to the results obtained with a base layout. The base layout has the same properties as the study case, except that the layout is not optimized but designed following a grid pattern.

|  | **Base layout** | **Optimized layout** | **Variation** |
|---|---|---|---|
| Obj [mEUR] | 4330.2 | 4364.5 | 0.8 % |
| $AEP_{pot}$ [GWh] | 2542.0 | 2560.8 | 0.7 % |
| Cable cost [mEUR] | 74.4 | 74.9 | 0.8 % |
| Anchors cost [mEUR] | 66.0 | 63.5 | -3.8 % |
| IAC length [km] | 99.4 | 100.2 | 0.8 % |
| Number of drilled anchors [-] | 38 | 34 | -10.5 % |
| Number of drag anchors [-] | 187 | 191 | 2.1 % |
| Electrical loss [GWh] | 4.2 | 4.7 | 11.8 % |
| Availability loss [GWh] | 6.3 | 6.0 | -3.7 % |
| OPEX [kEUR] | 699 | 705 | 0.8 % |

**Table 2.** Results for the base layout and for the optimized layout

From Table 2, it is seen that the objective function was increased by 0.8 % which amounts to 34.3 million euros (mEUR). The potential AEP was increased by close to 20 GWh by positioning the FOWTs in locations that reduce the wake effect. The AEP is one of the predominant driver of the NPV, since the evolution of the AEP has a similar trend as the objective function as shown in Fig. 7.





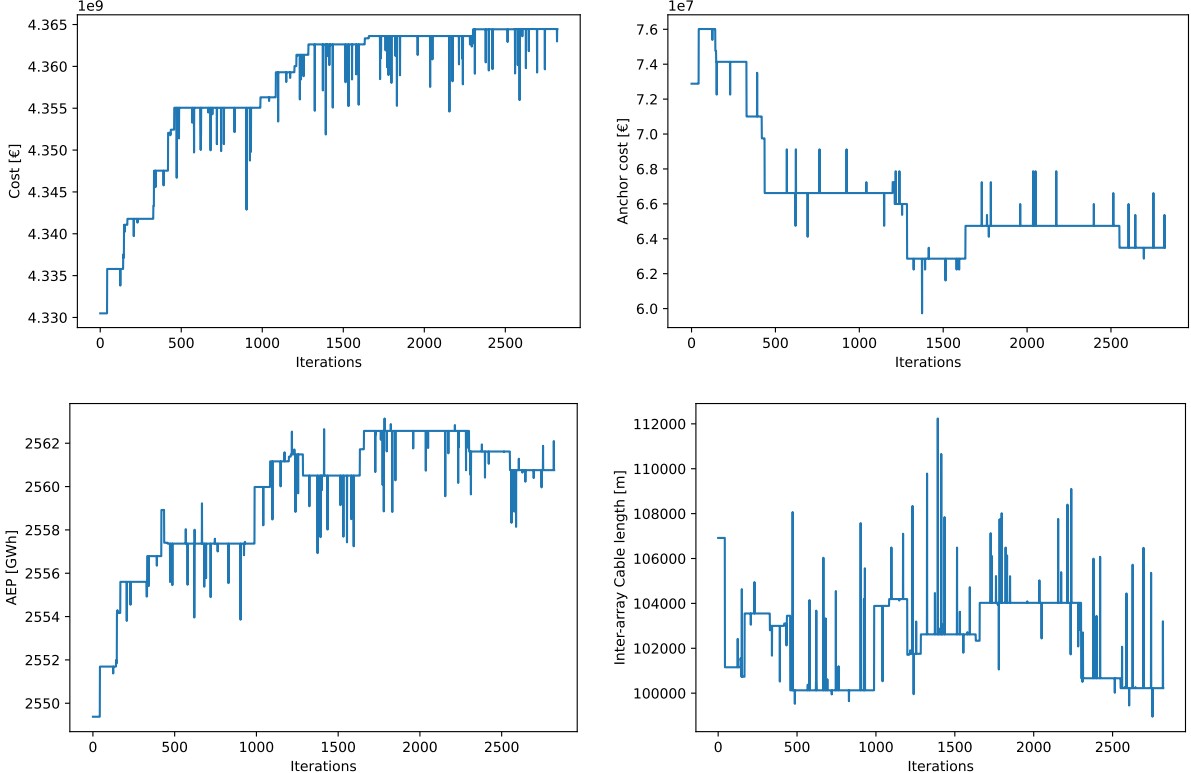

**Figure 7.** Evolution of parameters through the iterations of the optimizer

As for the CAPEX elements, the cable cost was slightly increased while the anchors cost was decreased by close to 4%. In that optimal scenario, the algorithm found out that moving the FOWTs out of the bedrock zone was providing a higher cost reduction than trying to reduce the IAC length. Indeed, when looking at Fig. 7, the anchors cost shows a descending trend while the cable cost - which is a linear function of the cable length - stays quite stable.

The slight decrease of the potential AEP around iteration 2300 goes together with a decrease of the cable cost and anchors cost. The algorithm ended with a scenario where the potential AEP is not fully maximized, since reducing slightly the CAPEX was leading to a better relative NPV.

As for the electrical losses and availability losses in Table 2, the electrical losses are slightly increased, because the total IAC length is higher and the production is also higher. However, the availability loss is reduced even though the production is increased. This is due to the combination of the following factors:

- There are more branches and therefore more turbines with no downstream turbines in the optimized layout than in the base layout, as shown in Fig. 8. The average number of turbines that are shut down due to a cable failure is lower in the optimized layout.

- The failure rate of each cable section depends on its length.





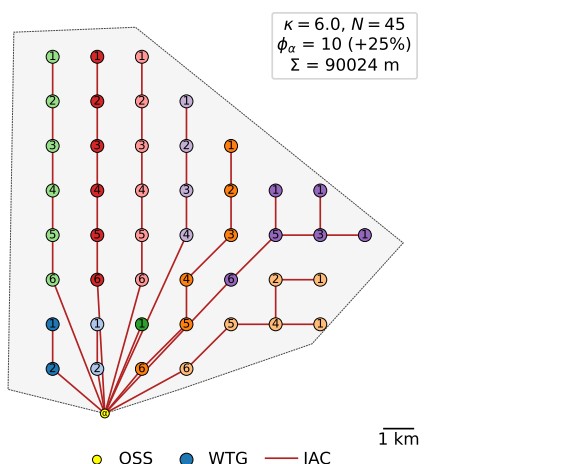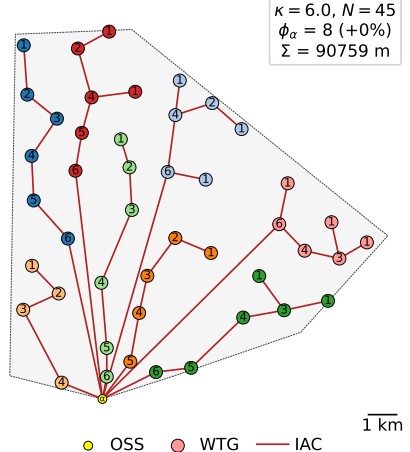

**Figure 8.** Grid-based layout (left) and optimized layout (right) and their associated optimized cable routings for Broadshore site

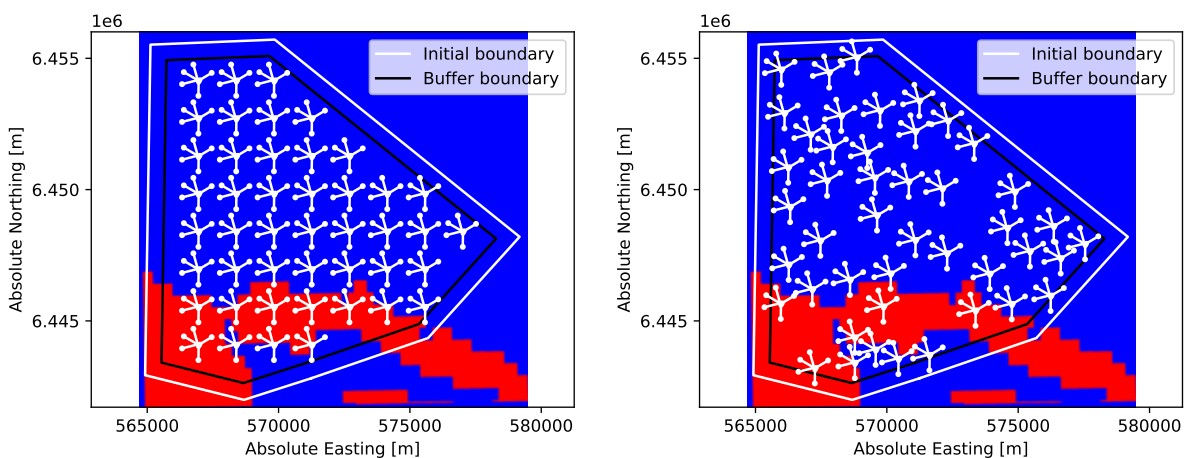

**Figure 9.** Grid-based layout (left) and optimized layout (right) showing WTGs with their mooring lines on the binarized bedrock map for Broadshore site

Compared to the base layout, the optimized layout shows that the turbines were moved as much as possible from the bedrock (red) zone, but because the anchors cost is not the only driver of the NPV, some anchors still fall in the bedrock, as shown in Fig. 9.

When looking at the base and the optimized layout in Fig. 9 from an aesthetic perspective, the grid-based layout looks more organized while the optimized seems to be messy, with the FOWT mooring line overlapping at the bottom of the site. However, the optimized layout satisfies the mooring line distance constraint of 80 m (for the optimized layout the minimum





distance between two mooring lines ended up being 81 m while for the base layout it was 312 m). Even if grid-based layouts
are sometimes preferred over irregular ones, optimized layouts bring a non-negligible gain of profit while at the same time
satisfying constraints to avoid technical or operational incidents.

To wrap up, it was found with the chosen technical and economic assumptions that the optimized NPV (objective function)
was increased by 34.5 mEUR compared to a grid-based layout. The top drivers of the objective function increase are listed in
Table 3, with their associated contribution. It is seen that the potential AEP is the main contributor, followed by the cost of
anchors and the availability gain. The components of the objective that ended up being 'worse' than in the grid-based layout
are all related to the cables. It can be said that with the cost assumptions chosen for that study case, the optimization of the
AEP and of the anchors is predominant over the IAC optimization.

| Parameter | Mathematical form | Contribution to the objective function |
|---|---|---|
| Potential AEP | $AEP_{pot}p_{kWh}a$ | +34.3 mEUR |
| Anchors cost | $C_{anchors}$ | +2.5 mEUR |
| Availability gain | $\eta_{avail}AEP_{pot}p_{kWh}a$ | +0.5 mEUR |
| OPEX | $C_{IAC,failure}a$ | -0.1 mEUR |
| Cable cost | $C_{cables}$ | -0.5 mEUR |
| Electrical loss | $\eta_{elec}AEP_{pot}p_{kWh}a$ | -0.9 mEUR |

**Table 3.** Ranking of the drivers of the objective function increase in the multi-parametric optimization

## 3.3 Sensitivity analysis

Because economic inputs are subject to a wide range of uncertainties, including changes in technology, energy prices, regula-
tions and consumer behavior, a sensitivity analysis is performed on the most uncertain economic inputs: the electricity price,
the anchors costs and the cable cost. Conducting sensitivity analysis allows to identify how changes in the economic inputs
can affect the performance and cost of a wind farm. This information can help to assess the risks associated with different
economic scenarios and to identify wind farm layout strategies to mitigate those risks.

### 3.3.1 Electricity price

In this sensitivity study, a simulation for a given set of parameters is run 5 times. Figures 10 and 11 show box plot of the dis-
tribution of the optimized parameters for different values of the electricity price. As expected, the mean value of the optimized
relative NPV grows linearly as the price of electricity increases, because an increased electricity price means more profit - if
the other parameters are fixed. Regarding the components of the objective function taken separately, they show a much wider
distribution around the median for a given set of parameters, than the objective function did. This is because the optimization
is based on a multi-parametric objective function, so different combinations of the parameters can lead to the same optimum.





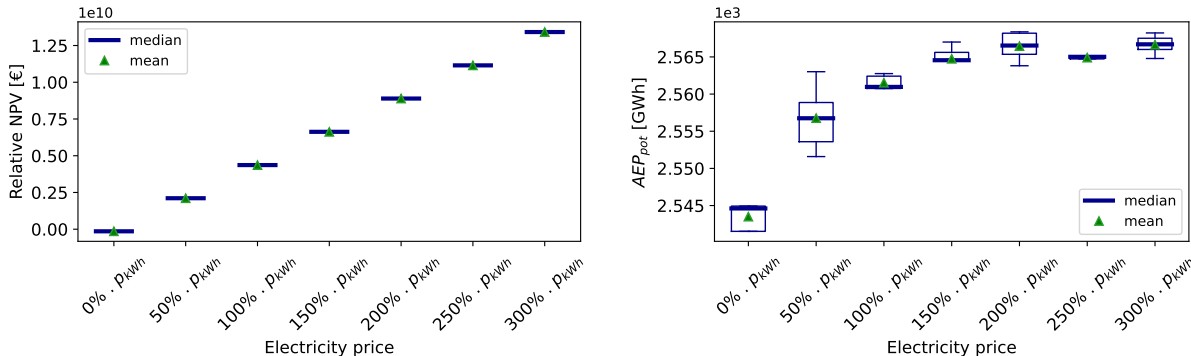

**Figure 10.** Results of the sensitivity analysis on the electricity price - relative NPV and potential AEP

The AEP evolution in Fig. 10 show an ascending trend as the price of electricity increases - while the CAPEX components (IAC and anchors) costs get worse (they augment). This is due to the fact that it becomes more and more worth it to move the FOWTs so that the AEP is increased, rather than placing them in zones where the associated CAPEX is low. Indeed, with the NPV as the objective function, the performance of the optimization is controlled by the trade-off between costs and AEP, which is defined by the assumed electricity price. On one side, for very high electricity prices, the CAPEX gets less important and the NPV objective approaches the AEP objective. For very low electricity prices, the AEP loses its importance and the optimization will be driven mainly by the CAPEX components.

It is also seen that when the electricity price reaches a certain level - here around $200\%.p_{kWh} = 211.2$ EUR/GWh - the AEP, the cable cost and the anchors cost seem to reach a threshold. This threshold stands for the limit above which it is not possible to increase the AEP any more - provided that the constraints are satisfied.

When looking closer at the CAPEX components - IAC and anchors - it is seen that for high electricity prices, the distribution of the total cable cost is quite wide, while it is not the case for the anchors' cost. This is due to the fact that the cable length is not only a driver of the CAPEX, but also of the electrical losses applied to the AEP and of the OPEX. This allows more liberty to the optimizer, making it possible to reach the same optimal objective function with different cable lengths. Different cable routings that have different total IAC lengths can reach to different availability losses / electrical losses and with a system of compensation, they can reach to the same objective function. As for the anchors, they only affect the CAPEX in this model, and their cost is driven by the positions of the FOWTs that make the anchors fall in bedrock zones or in sand zones: High anchors cost means a larger number of drilled anchors - located in bedrock zones - while low anchors cost means a larger number of drag anchors - located in sand zones.

The electrical loss in Fig. 12 shows a quite stable trend, varying by 0.6 GWh, between the maximal computed value and the minimal computed value. The electrical loss being a function of the production and of the cable length, an ascending curve would have been expected when the price of electricity increases, since both the cable length and the AEP grow as $p_{kWh}$ grows.





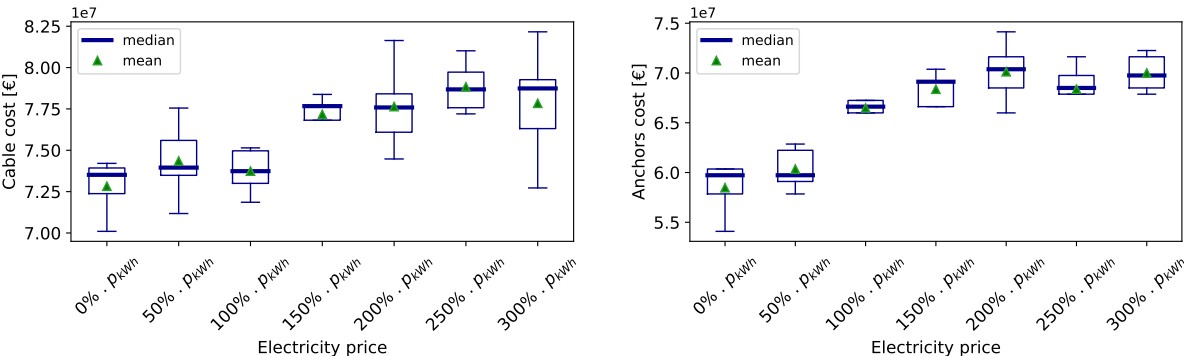

**Figure 11.** Results of the sensitivity analysis on the electricity price - CAPEX components (cable cost and anchors cost)

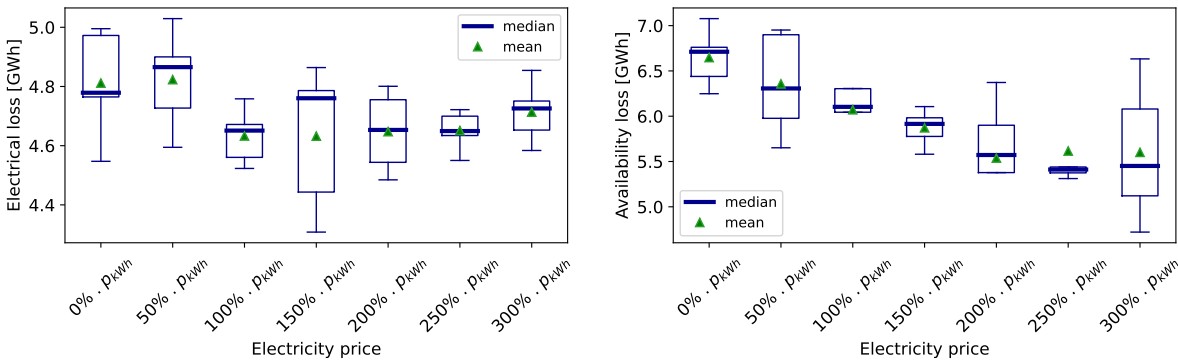

**Figure 12.** Results of the sensitivity analysis on the electricity price - Electrical and availability loss

However, the way the cable routing is designed also drives the electrical losses, and as it has been shown earlier (see Fig. 5), a higher number of strings and branches - and therefore less turbines per string - can lead to a lower electrical loss.

As for the availability losses, the trend is generally descending as the price of electricity increases, which is related to the cable layout and especially the average load of the wind farm. The average load of the wind farm is defined as the number of turbines connected downstream of a given turbine, averaged across the whole site. In other words, the average load can be seen as the number of turbines affected by the failure of a cable section. In this study, the average load of the wind farm decreases as the price of electricity grows. Therefore, the availability is optimized by playing with the cable routing design and it is possible to make it decreasing even if the production is increased.

To investigate the impact of the electricity price on the optimized layout, a density heat-map is generated for different levels of $p_{kWh}$. It allows to find out if the optimization process delivers representative layouts with recurring trends in terms of optimal FOWTs positions. The heat-map is generated by estimating the probability density function of the set of optimized coordinates of the FOWTs. It is based on the Kernel Density Estimation (KDE) method, which is a non-parametric way of

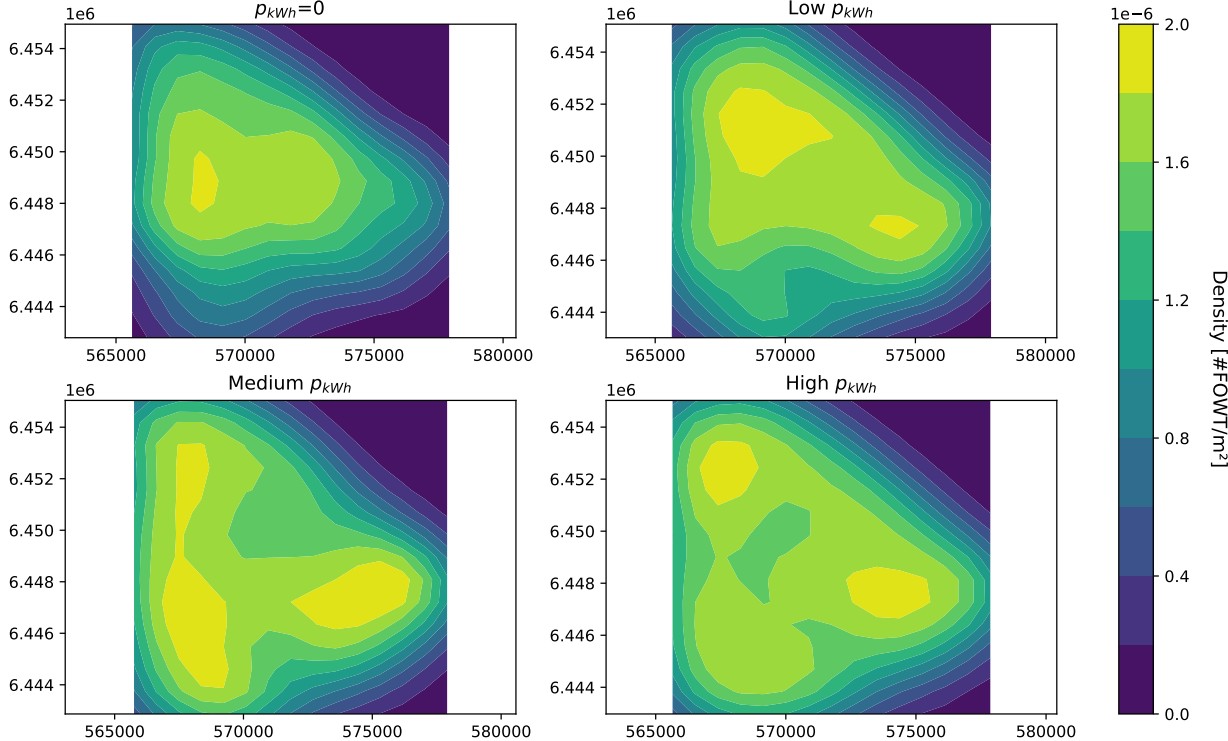

**Figure 13.** Density heat-maps of FOWT positions, for different electricity price levels

*x-axis: Absolute Easting [m]    ,    y-axis: Absolute Northing [m]*

estimating the PDF of a random variable. In this study, for each electricity price level, 5 optimal layouts have been generated. The five layout are merged together and the Gaussian KDE is computed for 0, low (50 % $p_{kWh}$), medium (100 % $p_{kWh}$) and high (250 % $p_{kWh}$) electricity prices, as shown in Fig. 13.

For $p_{kWh} = 0$ in the top left hand corner of Fig. 13, the optimized layout shows a trend of centering the turbines in the center of the site. In that unrealistic scenario, the AEP is totally neglected, and the anchors and IAC are the only components of the

objective function. Therefore, the optimizer gathers the FOWTs together to the south of the site where the OSS is located to reduce as much as possible the IAC cost, but at the same time it avoids the bedrock zone at the South, to minimize the anchors cost. For a low price of electricity, then the AEP is taken into account but it is not predominant, since it is seen that the south bedrock zone is still avoided. As the price of electricity increases, the turbines get positioned more and more evenly all over the site, with a density that tends to converge to a constant density for all locations. Overall, the electricity price affects directly

how predominant the AEP is compared to the CAPEX elements, and therefore to what extent it is worth it in terms of revenue to spread the turbines across the site to reduce the wake effect.



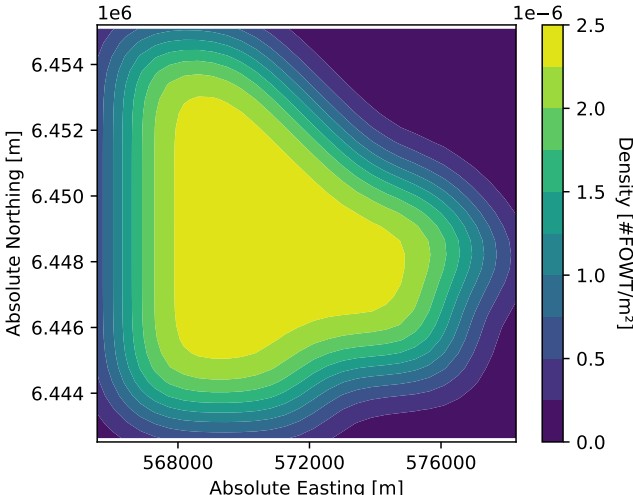

**Figure 14.** Density heat-maps of the grid-based layout

To assess the variation of the optimized layout compared to the grid-base layout generated in Sect. 3.2, a correlation coefficient between the densities is calculated in Table 4. This is done by correlating the z values of each of the plots in Fig. 13 together with Fig. 14.

The cases where the electricity price is low have a high correlation coefficient, because the density heat-maps show a quite uniform distribution across the site, just like for the grid-based density heat-map. Then the case $p_{kWh}300$ % also has a high correlation coefficient since - as it has been said earlier - the turbines are distributed rather evenly on the site, which is similar to the grid-based layout. As for the in-between electricity price values, the correlation is at its lowest because the layouts show an uneven distribution of the FOWTs on the site.

| Sensitivity [%] | Correlation [%] |
|---|---|
| 0 | 80.9 |
| 50 | 81.9 |
| 100 | 71.3 |
| 150 | 78.5 |
| 200 | 70.5 |
| 250 | 75.9 |
| 300 | 81.5 |

**Table 4.** Table of correlation with the base layout for the electricity price sensitivity analysis





### 3.3.2 Anchors cost

To stay consistent with the logic of the algorithm, being that it is more expensive to install drilled anchors than drag anchors, the cost of the anchors are incremented together:

$$C_{Drag\ anchor}^{sens} \in C_{Drag\ anchor}[0\,\%, 50\,\%, 100\,\%, 150\,\%, 200\,\%] \tag{17}$$

$$C_{Drilled\ anchor}^{sens} \in C_{Drilled\ anchor}[0\,\%, 50\,\%, 100\,\%, 150\,\%, 200\,\%] \tag{18}$$

It is important to make the difference between the two types of anchors vary from a sensitivity to another. Indeed, keeping the same difference of cost while making the cost of anchors increase would result in a single scenario, making the increased cost of anchors act as a fixed cost. Here, it is the impact of choosing a type of anchor over another that drives the behavior of the optimization.

In a similar way as it was stated in the cable cost sensibility analysis, the relative NPV decreases as the anchors' cost increases, because one of the CAPEX terms gets more expensive. Additionally, a variation of the anchors cost seems to have a rather small impact on the relative NPV compared to a variation of the electricity price: the relative NPV varies by 140 mEUR over the whole range of sensitivities. Apart from that, the conclusions drawn from the cable cost sensitivity analysis can be applied for the anchors as well: Both the AEP and the IAC cables become less predominant as the cost of anchors increases. For a high unit cost of anchors, the AEP gets lower and the cable length increases because the gain of profit is higher when focusing the optimization on the anchors. Unlike the AEP and the IAC, the anchors variable is a discrete variable, meaning that the number of drag anchors and drilled anchors are bounded and correlated, as shown in Equations (19) and (20).

$$n_{Drag\ anchor} \in [0, n_{mooring}N] \tag{19}$$

$$n_{Drilled\ anchor} \in [n_{mooring}N - n_{Drag\ anchor}] \tag{20}$$

Knowing Equations 19 and 20, a threshold is expected, where $n_{Drag\ anchor} = n_{mooring}N$ and $n_{Drilled\ anchor} = 0$ - this would happen for extreme prices of anchors. In this study, no threshold has been reached, because the anchors price hasn't been set high enough to dominate the whole multi-parametric optimization. It could also be that the threshold is not reachable within the problem's boundaries - especially the geometrical distance constraint for the mooring lines can be a limiting factor.

*Similar figures to the ones presented in Sect. 3.3.1 are available in Appendix B2.*

### 3.3.3 Unit cable cost

When increasing the unit cable cost, the relative NPV drops. This result suggests that the increase of cable cost between each sensitivity simulation leads to a larger decrease of the relative NPV than the possible increase of relative NPV through its optimization.

Both the potential AEP and the cable length decrease as the unit cable cost increases. The reason for that is that when the cable cost per unit length gets higher, then minimizing the cable length becomes the priority over the maximization of the AEP (and similarly for the minimization of the anchors' cost), because it leads to a higher increase of the objective function.

*Similar figures to the ones presented in Sect. 3.3.1 are available in Appendix B3.*



# 4 Conclusions

In this project, a techno-economic multi-parametric layout optimization model has been developed for FOW. The lack of models available for FOW accounting for both technical and economic aspects makes the present work a state-of-the-art tool - ready for further developments.

FOWFs face unique challenges compared to BOWFs, with a higher complexity, and more constraints. Therefore, floating specific wind farm layout optimization is crucial to ensure that floating projects are economically viable and technically reliable. Optimization - whether it is related to the design of the components or on the layout - behaves as a vector to help floating projects to be approved. Today, the majority of the offshore projects have a grid-based layout - but the tool developed in this study can provide economic indicators based on scientific models that prove how attractive it is to resort to layout optimization.

In the present work, the relative NPV has been used as the objective of the optimization. Using the relative NPV has allowed to only include relevant parameters that vary with the actual FOWF layout: the potential AEP, the IAC routing, the type of anchors and the losses associated to the IAC. Maximizing such a multi-objective function that combines the capital investment together with the operational costs and the energy production profit has proven that it's possible to find the best balance between all the costs elements according to the specificity of the site, of the wind farm, of the economic inputs etc.

A sensitivity analysis has been carried out to help identify which cost inputs have the greatest impact on the model output, thereby allowing decision-makers to focus their efforts on addressing the most important uncertainties. In this study, it was found that the electricity price fluctuations affect the most the final relative NPV: it was proven that even small deviations from the electricity price forecast can have significant financial implications. Nevertheless, this conclusion should be treated cautiously as it is highly dependent on the wind farm properties, the available wind resource, the cost inputs etc. Overall, the sensitivity analysis can help decision-makers identify opportunities for cost reduction according to the project's specifications and assumptions.



## Appendix A: Inputs

### A1 Technical inputs

| Category | Parameters | Value |
|---|---|---|
| **Wind farm** | N [-] | 45 |
| | $\mathcal{A}$ [km$^2$] | 134 |
| | $n_y$ [years] | 20 |
| **WTG** | $P_{rated}$ [MW] | 11.3 |
| | $D$ [m] | 200 |
| | Cut-in wind speed [m/s] | 3 |
| | Cut-out wind speed [m/s] | 30 |
| **Floating** | $n_{mooring}$ [-] | 5 |
| | Mooring footprint [m] | $\in [521,616]$ |
| | $d_{min}^{mooring}$ [m] | 80 |
| | Floating platform [-] | Semi-submersible |
| | Floating platform draft [m] | 20 |
| | $\Phi$ [rad] | $\pi/3$ |
| **BoP** | IAC capacity [MW] | 71 |
| | $\kappa$ [-] | 6 |
| | Resistance [$\Omega$/km] | 0.03 |
| | Voltage [kV] | 66 |
| | Failure rate [Failures/y] | 0.0094 |
| | Time to repair IAC [hours] | 1080 |
| | $n_{OSS}$ [-] | 1 |
| **Site** | $z_{depth}$ [m] | 90 |
| | $z_0$ [m] | 2E-4 |
| | TI | 0.1 |

**Table A1.** Technical inputs for Broadshore site



## A2  Economic inputs

| Category | Parameters | Value |
|---|---|---|
| | $p_{kWh}$ [EUR/MWh] | 105.6 |
| | $r$ [%] | 1.74 |
| Economic | $C_{IAC}$ [EUR/m] | 748 |
| | $C_{Drilled\ anchor}$ [kEUR/unit] | 814 |
| | $C_{Drag\ anchor}$ [kEUR/unit] | 187 |

**Table A2.** Economic inputs for Broadshore site

## Appendix B:  Sensitivity study

### B1  Price of electricity

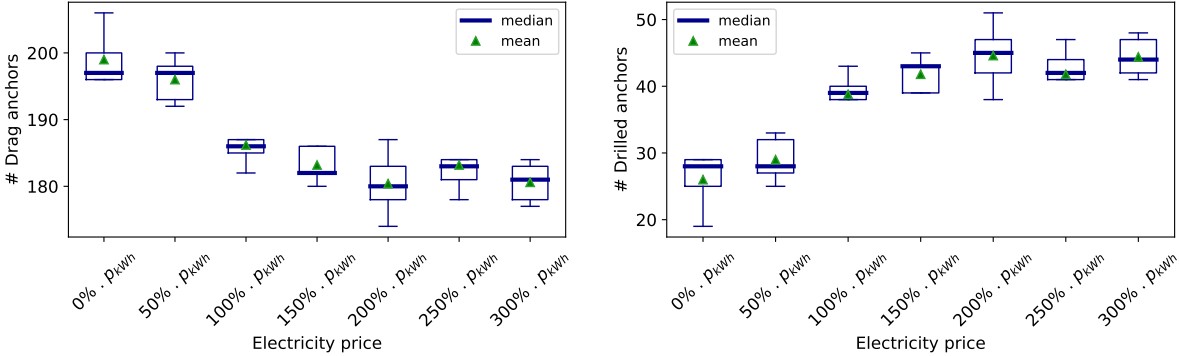

**Figure B1.** Results of the sensitivity analysis on the electricity price - Number of drag and drilled anchors



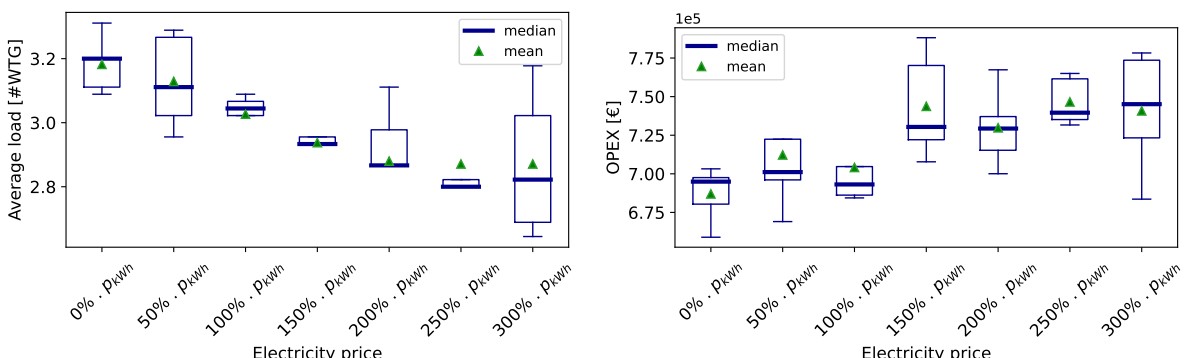

**Figure B2.** Results of the sensitivity analysis on the electricity price - Average load and OPEX

## B2    Anchors' cost

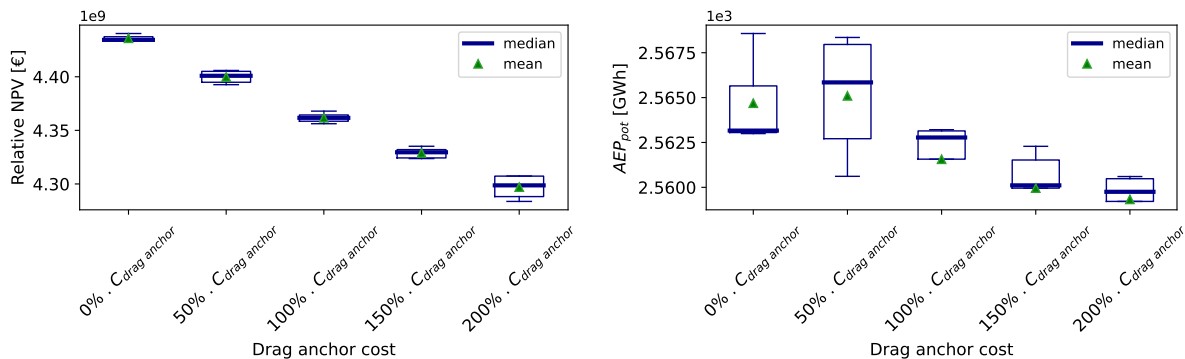

**Figure B3.** Results of the sensitivity analysis on the anchors cost - relative NPV and potential AEP

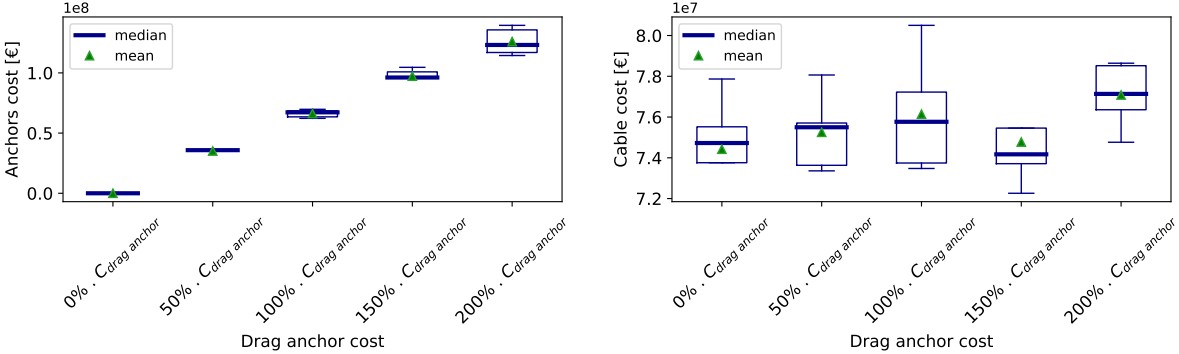

**Figure B4.** Results of the sensitivity analysis on the anchors cost - Anchors costs and cable cost




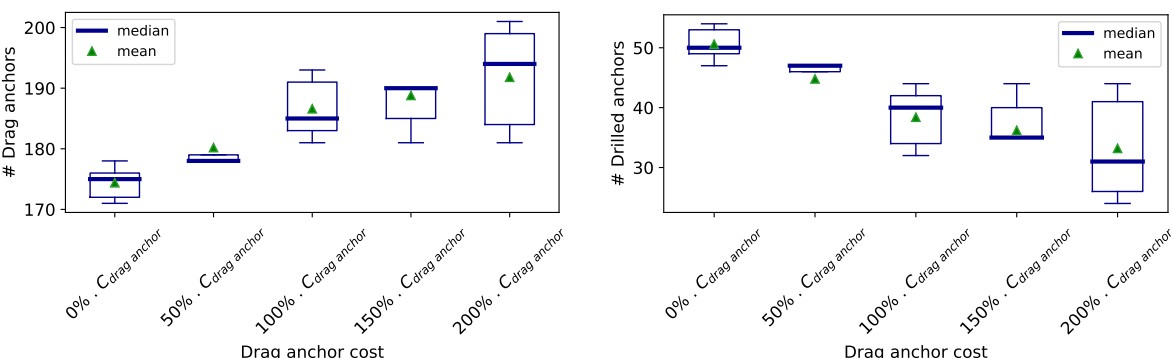

**Figure B5.** Results of the sensitivity analysis on the anchors cost - Number of drag and drilled anchors

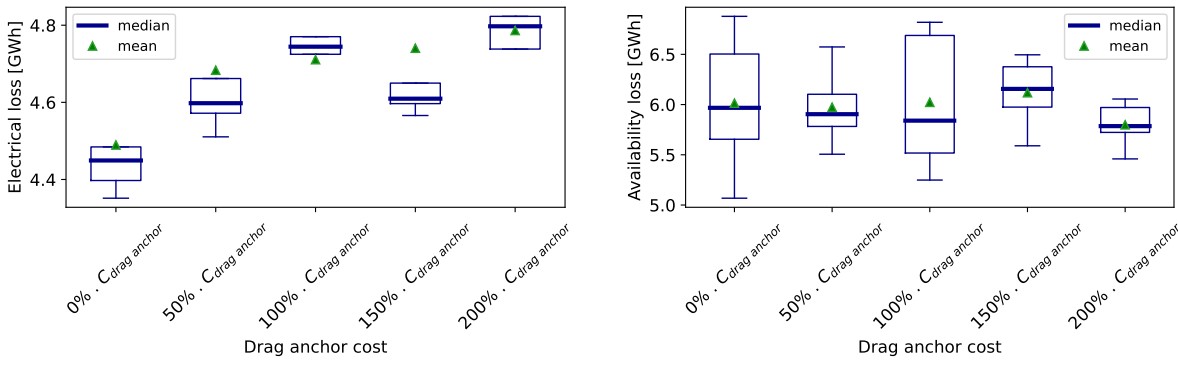

**Figure B6.** Results of the sensitivity analysis on the anchors cost - Electrical and availability losses

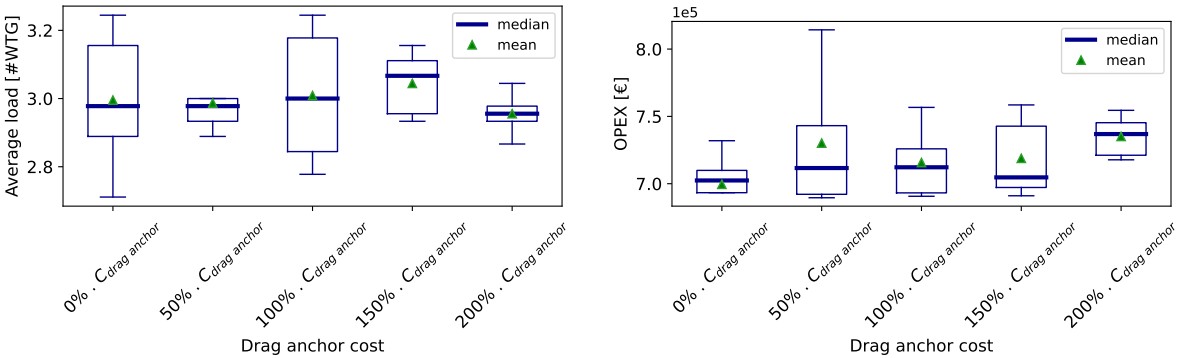

**Figure B7.** Results of the sensitivity analysis on the anchors cost - Average load and OPEX



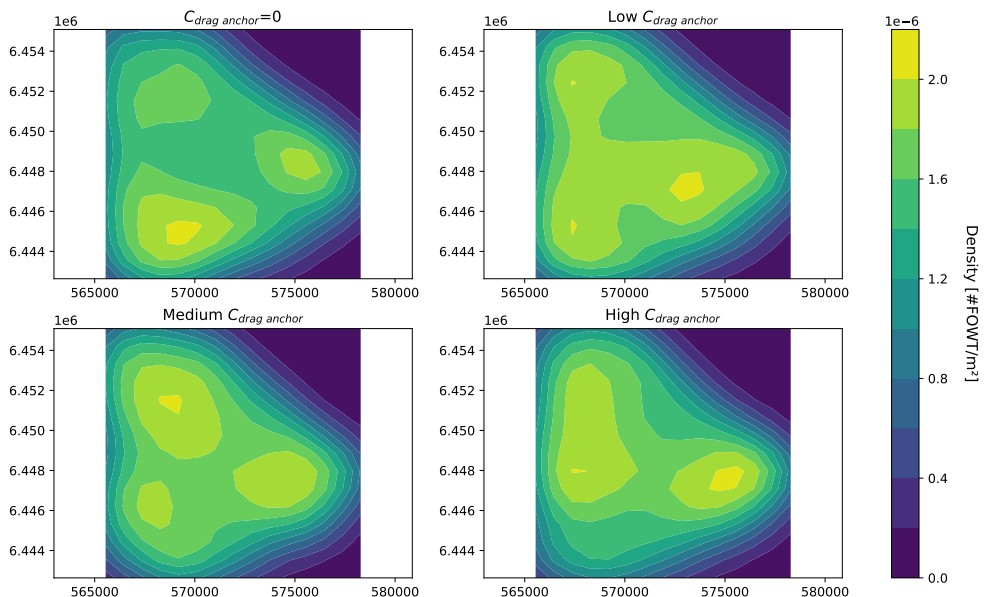

**Figure B8.** Density heat-maps of FOWT positions, for different anchor cost levels

*x-axis: Absolute Easting [m]    ,    y-axis: Absolute Northing [m]*

## B3  Unit cable cost

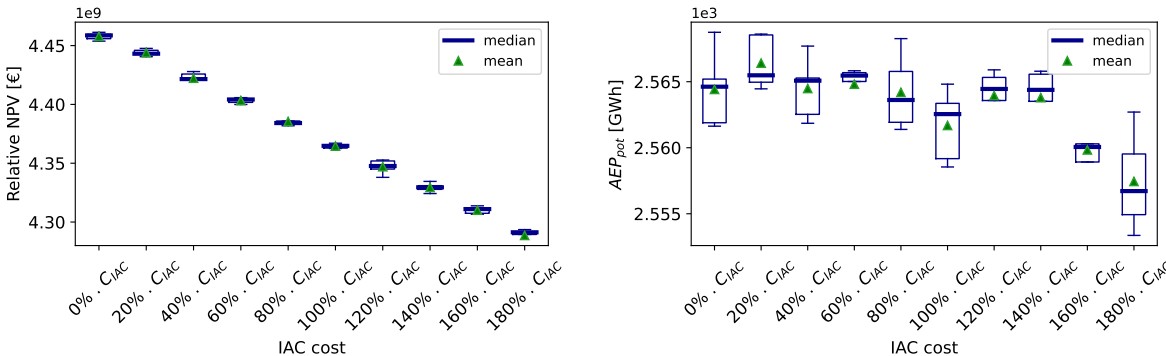

**Figure B9.** Results of the sensitivity analysis on the unit cable cost - relative NPV and potential AEP




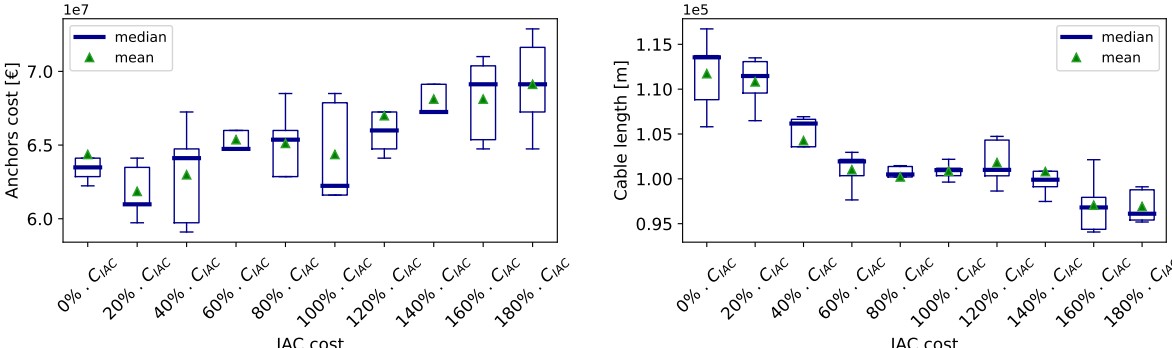

**Figure B10.** Results of the sensitivity analysis on the unit cable cost - Anchors costs and cable length

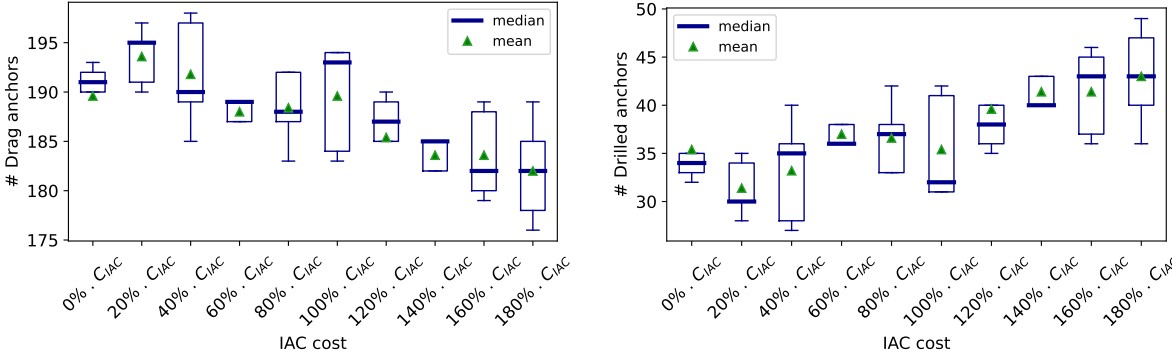

**Figure B11.** Results of the sensitivity analysis on the unit cable cost - Number of drag and drilled anchors

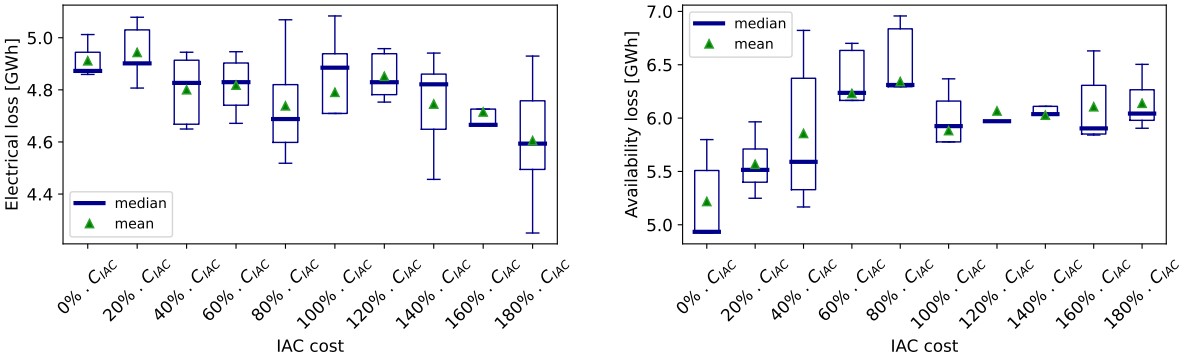

**Figure B12.** Results of the sensitivity analysis on the unit cable cost - Electrical and availability losses





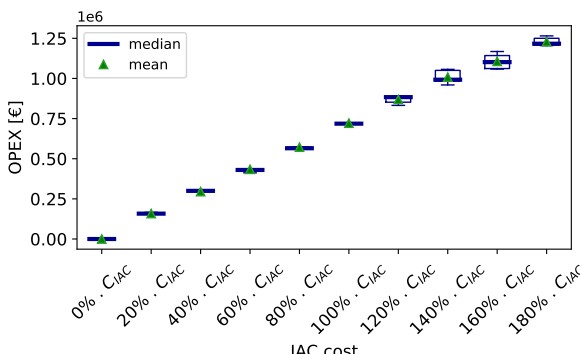

**Figure B13.** Results of the sensitivity analysis on the unit cable cost - OPEX

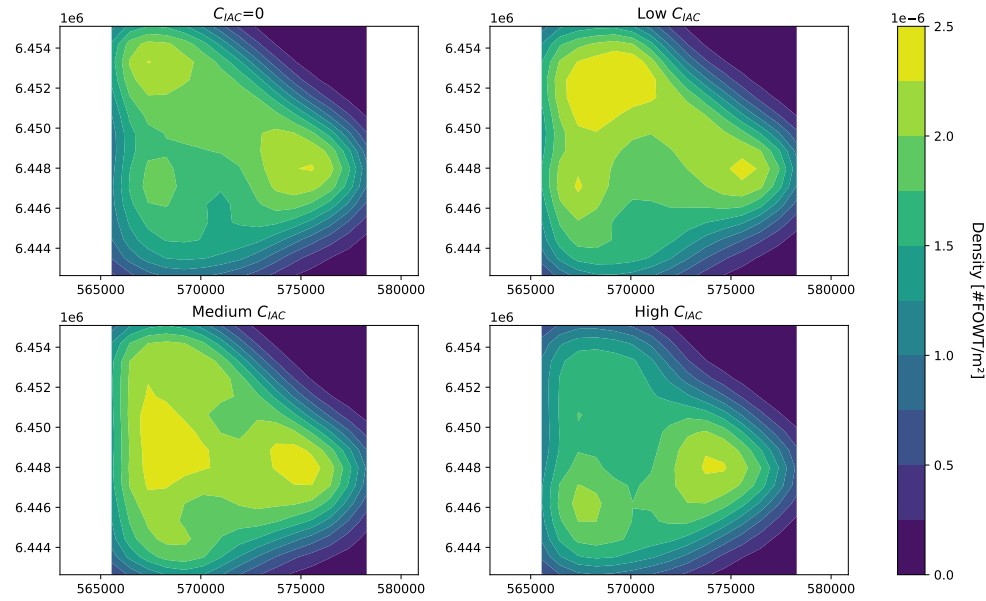

**Figure B14.** Density heat-maps of FOWT positions, for different unit cable cost levels

*x-axis: Absolute Easting [m]    ,    y-axis: Absolute Northing [m]*



*Author contributions.* I. Bayati, T.H. Snedker and K. Dykes proposed, supported and reviewed the present work. A.I. Hietanen proposed the assumptions and methodology, programmed the models, created the simulations, results and conclusions. A.I. Hietanen also edited the present paper.

*Competing interests.* Some authors are members of the editorial board of WES journal.

*Acknowledgements.* A.I. Hietanen would like to thank her supervisors for their guidance, expertise and support throughout the project. In addition, she would like to acknowledge PEAK Wind for accommodating the creation of a scientific hub for development to benefit the Wind Industry.





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
