# Peer review of "A novel techno-economical layout optimization tool for floating wind farm design"

_Wind Energy Science, 2023_

## Referee Comment (RC2)

The manuscript does a very good job presenting a novel method for floating offshore wind farms design and optimization. This is a novel and relevant work, which sheds light on different aspects of a multidisciplinary problem. The paper is in general clear and well-structured. The method is detailed and fills many gaps between fixed bottom and floating wind farm designs.

General comments:

- The paper is lacking details on how the AEP was calculated. Which wake model was used and how it was set?
- Is there a plan to make the methods and the code used open-source and publicly available?
- The conclusion should be covering all aspects of the paper and can stand as a standalone text, which explains the paper objectives, methods used and goes over the results. This is currently not the case.

Specific comments:

- Line  24: typo
- Figure1: Use the turbine diameters to normalize the x-y axis (Please do the same for similar Figures such as Figure 9). I also recommend adding a grid to both subplots.
- d mooing minimum, is not defined till the case study section. Why is the value of 80 meters is taken into account? Is it a case specific value? The API standard specifies 20m distance between mooring lines is enough.
- Equation 7: Clarify how AEP losses were calculated.
- Line 117: Did you include the effect of the FOWTs while calculating the AEP? If not you have to clearly state it in the text.
- Line118: What is the maximum displacement allowed by the mooring system design in this work? At which wind direction does it happen?
- Figure2: Please add a legend to the subplot on the right. Also add a title to the color bar in the subplot on the left and indicate the units used (m\km?)
- Line 156: You state that the algorithm produces better results than many other heuristics. Is there any literature review to support you statement?
- Figure 3: Some parameters in the legend were never clarified or mentioned in the text.
- Equation 11: This is used to calculated total length after the routing? I feel this function can be only applied between two wind turbines so N is a bit confusing. Can you clarify?
- In line 220 explaining figure 5. The number of OSS changes. Is this an input to the optimization or is it an output of the optimization? If it is an input do you think it affects your results? Why did you decide on two different values for OSS?
- Figure 9: Why is the gridded layout only within the buffer boundary? This means that your turbines for the gridded layout is more closer to each other which will affect the AEP and increase the losses of this layout?

---

## Author Response (AR1)

**RESPONSE TO REVIEW 1 (RC1, 26 Oct. 2023)**

Thank you for your constructive review of the manuscript. I am pleased to inform you that I have carefully addressed each of your comments to enhance the scientific quality of the research article. Specifically:

**Literature Review:**
I have incorporated a comprehensive literature review in the introduction section, outlining existing research in the field of layout optimization. This addition highlights the novelty of our work and underscores the main differences between the developed optimization model and previously published models for both bottom-fixed and floating offshore wind farms.

**AEP Methodology:**
I have included a short description of the methodology employed to determine the AEP and wake losses. Additionally, I have provided an equation illustrating the relationship between AEP and the two efficiencies—electrical losses and availability losses, addressing the factors impacting the model.

**Optimization Algorithm:**
Chapter 2.3 now features an explanation of the applied optimization algorithm, including its methodology.

We are grateful for your suggestion regarding further constraints in the optimization tool. Your insight into including elements such as shipping routes, environmental protection areas, fishery zones, etc., is highly relevant and aligns with our consideration for future developments in the research. We will certainly explore these aspects to enrich the comprehensiveness of the wind farm layout development.

**RESPONSE TO REVIEW 2 (RC2, 24 Nov. 2023)**

Thank you for your thorough evaluation and constructive feedback on our manuscript. I have incorporated the specific comments you provided into the manuscript. For the general comments you raised:

**AEP Methodology:**
I have included a description of the AEP methodology in the manuscript. The wake model used is the Bastankhah and Porté-Agel wake model, chosen for its balance between computational efficiency and accurate far wake modelling. The potential AEP was computed using the PyWake Python library.

**Code Availability:**
While I appreciate the importance of open-source methods, at this moment, the code is not planned to be made publicly available.

**Conclusion Modification:**
I have revised the conclusion to comprehensively cover all aspects of the paper, including the objectives, methods, and an overview of the results.

**LIST OF ALL RELEVANT CHANGES**

- Literature review added.
- Figure 1: x-y axis normalized.
- Elaboration of the AEP calculation methodology (tools, wake loss model).
- Equation AEP-efficiencies added. Clarification of how the losses were computed.
- Description of the process and methodology of the optimization algorithm.
- Mention of the intentional omission of the dynamic behavior of the mooring system.
- Figure 2: Legend added on the right subplot. Color bar title and units added for the left subplot.
- Figure 3: Clarification of the legend.
- Equation 11: Adjustment of the equation for two entities.
- Elaboration on the number of OSS chosen to compute Figures 5 and 6.
- Additional remarks: elaboration on the mooring distance constraint values + comment on the grid layout used in Section 3.
- Adjustment of the conclusion with the paper's objectives, the methods used and the main results.